# RIGID PROTEIN-PROTEIN DOCKING VIA EQUIVARIANT ELLIPTIC-PARABOLOID INTERFACE PREDICTION

**Ziyang Yu**[1]  **Wenbing Huang**[3,4,*] **Yang Liu**[1,2,*]

[1]Department of Computer Science, Tsinghua University
[2]Institute for AI Industry Research (AIR), Tsinghua University
[3]Gaoling School of Artificial Intelligence, Renmin University of China
[4]Beijing Key Laboratory of Big Data Management and Analysis Methods, Beijing, China
`yuziyang20@mails.tsinghua.edu.cn, hwenbing@126.com, liuyang2011@tsinghua.edu.cn`

## ABSTRACT

The study of rigid protein-protein docking plays an essential role in a variety of tasks such as drug design and protein engineering. Recently, several learning-based methods have been proposed, exhibiting much faster docking speed than those computational methods. In this paper, we propose a novel learning-based method called ElliDock, which predicts an elliptic paraboloid to represent the protein-protein docking interface. To be specific, our model estimates elliptic paraboloid interfaces for the two input proteins respectively, and obtains the roto-translation transformation for docking by making two interfaces coincide. By its design, ElliDock is independently equivariant with respect to arbitrary rotations/translations of the proteins, which is an indispensable property to ensure the generalization of the docking process. Experimental evaluations show that ElliDock achieves the fastest inference time among all compared methods and is strongly competitive with current state-of-the-art learning-based models such as DiffDock-PP and Multimer particularly for antibody-antigen docking.

## 1 INTRODUCTION

Proteins realize their biological functions through molecular binding sites, which depend on the complementary in structure and biochemical properties around key regions of protein complexes. Studying the mechanism of protein interaction is of great significance for drug design and protein engineering. In this work, we aim to tackle *rigid protein-protein docking*, which means we only target at learning rigid body transformation from two unbound structures to their docked poses, keeping the conformation of the protein itself unchanged during docking.

Traditional docking softwares (Chen et al., 2003; Venkatraman et al., 2009; De Vries et al., 2010; Moal et al., 2013; de Vries et al., 2015; Sunny & Jayaraj, 2021) mainly follow this general paradigm: first sampling numerous candidate poses and then updating the structures with higher scores based on a well-designed energy model. These methods are highly computationally expensive, which may be unbearable to dock millions of complex structures in practice.

Recently, learning-based methods (Gainza et al., 2020; Ganea et al., 2021; Evans et al., 2021; Ketata et al., 2023; Luo et al., 2023) have been

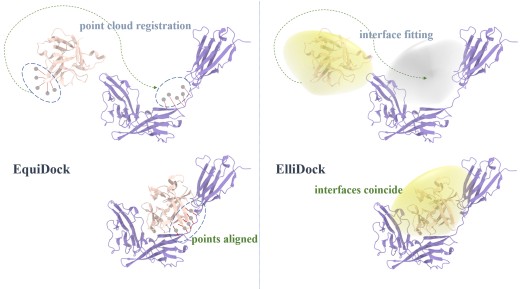

Figure 1: EquiDock: point cloud registration vs ElliDock: interface fitting.

applied to the task. EquiDock (Ganea et al., 2021) is a regression-based model that tries to predict the functional pocket of each of the receptor and the ligand proteins, and apply point cloud registration to obtain docking transformation. However, we believe that rigid protein-protein docking task is not

---

[*]Corresponding authors: Wenbing Huang, Yang Liu.

entirely equivalent to local point cloud registration for two reasons: **1.** the docking transformation is unable to be derived uniquely if the registered point cloud is ill-posed and degenerated to be, for example, within 2-dimensional planes or even 1-dimensional lines; **2.** it will cause miss-alignment if the registered point clouds of the two proteins are not exactly the respective docking pockets. Another related work is MaSIF (Gainza et al., 2020), which is surface-based and designed to map the geometric and chemical features of local protein regions into fingerprints. MaSIF can well demonstrate the biochemical properties of the protein monomer surface (e.g., hydrophilicity) with a lack of information interaction during protein-protein docking, which may hinder the discovery of protein-specific binding sites. More recently, xTrimoDock (Luo et al., 2023) leverages a cross-modal representation learning to enhance the representation ability. Since the model is carefully designed based on specific regions (e.g., heavy chains) on the antibody, it lacks generalizability across the entire protein domain. Another diffusion-based generative model named DiffDock-PP (Ketata et al., 2023) is proposed to obtain high-quality docked complex structures, yet at the expense of much longer inference time and careful tuning of many sensitive hyper-parameters.

**Contribution.** We propose ElliDock, a novel method tailored for rigid protein-protein docking based on global interface fitting. Our model first predicts a pair of elliptical paraboloids with the same shape as the binding interfaces, based on the global information of the two proteins. Then we obtain the roto-translation for docking by calculating the transformation that makes two paraboloids coincide. Nicely, our ElliDock is independently equivariant with respect to arbitrary rotations/translations of the proteins, which is an indispensable property to ensure the generalization of the docking process. Meanwhile, by constraining predicted interfaces to spatially separate nodes of the receptor and the ligand, ElliDock manages to avoid steric clashes, which is a hard-core problem faced by some of the previous works. Figure 1 illustrates the difference between EquiDock and our ElliDock.

We conduct experiments on both the Docking Benchmark database (DB5.5) (Vreven et al., 2015) and the Structural Antibody Database (SAbDab) (Dunbar et al., 2014), mainly exploring the performance of our model in heterodimers like antibody-antigen complexes. Experimentally, ElliDock is strongly competitive with current state-of-the-art learning-based models such as DiffDock-PP (Ketata et al., 2023) and Multimer (Evans et al., 2021) particularly for antibody-antigen docking, with significantly decreased inference time compared with all docking baselines.

## 2 RELATED WORK

**Equivariant Graph Neural Networks (EGNNs).** EGNNs are tailored for data lying in the Euclidean space (e.g., the coordinate of an atom), which is unable for plain GNNs to achieve equivariance from different coordinate transformation (Satorras et al., 2021; Keriven & Peyré, 2019). EGNNs propose a solution by designing equivariant layers and utilizing the equivariant information in the graph. Therefore they are suitable for tasks such as molecular dynamical systems, interaction affinity prediction and protein structure generation (Fuchs et al., 2020; Ganea et al., 2021; Jing et al., 2021; Hutchinson et al., 2021; Wu et al., 2021; Kong et al., 2022; Jiao et al., 2022). Due to the particularity of the protein-protein docking task (with a random unbound structure, the interaction between proteins cannot be effectively constructed), we specially design an pairwise $SE(3)$-equivariance message passing network with global-level interaction, which sets it apart from the previous work.

**Protein-Protein Docking.** Protein complexes data acquired in the laboratory is not only expensive but also inefficient to produce, so we resort to finding efficient computational methods to discover protein-protein docking patterns (Vakser, 2014). Traditional docking methods (Chen et al., 2003; Venkatraman et al., 2009; De Vries et al., 2010; Satorras et al., 2021; Biesiada et al., 2011; Schindler et al., 2017; Sunny & Jayaraj, 2021) share a similar pattern with relatively high computational cost: sample millions of candidate complex poses at first, rank them with a well-designed scoring function, then update the complex structures among the top-ranked candidates based on an energy model. Deep learning methods are recently proposed to tackle protein-protein docking task based on molecular dynamics (Desta et al., 2020; Ghani et al., 2021), surface property deciphering (Gainza et al., 2020), keypoints alignment (Ganea et al., 2021), protein structure prediction (Jumper et al., 2021; Evans et al., 2021), multiple sequence alignment (Bryant et al., 2022), diffusion (Ketata et al., 2023) or cross-modal representation learning (Luo et al., 2023). Different from prior work, we propose a novel method that converts rigid protein docking to predicting binding interfaces as a pair of elliptic paraboloids and calculating the transformation based on interface fitting.

## 3 METHOD

As defined in Section 1, the rigid protein-protein task aims to predict a SE(3) transformation (rotation and translation) of the unbound structures of an interacting protein pair to get the docked complex. To address this, we first develop a pairwise-independent SE(3)-equivariant GNN, named EPIT, to characterize both the intra-protein and inter-protein interactions in the complex. Then, upon the invariant and equivariant outputs of EPIT, we derive the SE(3) transformation via elliptic paraboloid interface prediction and alignment.

### 3.1 PROTEIN REPRESENTATION

We model a protein as a graph by regarding each residue as a node, and denote $\mathcal{G}_1 = (\mathcal{V}_1, \mathcal{E}_1)$, $\mathcal{G}_2 = (\mathcal{V}_2, \mathcal{E}_2)$ as the ligand graph and the receptor graph, respectively. We associate each node $v_i$ with a tuple $(\boldsymbol{h}_i, \vec{\boldsymbol{x}}_i)$ where $\boldsymbol{h}_i \in \mathbb{R}^H$ denotes the SE(3)-invariant feature and $\vec{\boldsymbol{x}}_i \in \mathbb{R}^3$ is the 3D coordinate of the $\alpha$-carbon atom in residue $i$. The collection of all node representations yields $\boldsymbol{H}_1 \in \mathbb{R}^{N_1 \times H}, \vec{\boldsymbol{X}}_1 \in \mathbb{R}^{N_1 \times 3}$ for the ligand graph $\mathcal{G}_1$, and $\boldsymbol{H}_2 \in \mathbb{R}^{N_2 \times H}, \vec{\boldsymbol{X}}_2 \in \mathbb{R}^{N_2 \times 3}$ for the receptor graph $\mathcal{G}_2$. The edges are constructed in the k-NN manner, that is, selecting $k$ nearest nodes in the Euclidean distance from the same protein as the neighbors for each node $v_i$. More details of the graph representation are provided in Appendix.

**Task Formulation** We fix the conformation of the receptor $\mathcal{G}_2$, and predict a rotation matrix $\boldsymbol{Q} \in \mathrm{SO}(3)$ and a translation vector $\vec{\boldsymbol{t}} \in \mathbb{R}^3$, such that the transformation of the ligand $\mathcal{G}_1$, namely $\vec{\boldsymbol{X}}_1^* = \vec{\boldsymbol{X}}_1 \boldsymbol{Q} + \vec{\boldsymbol{t}}$ is at the docking position to the receptor.

### 3.2 EQUIVARIANT POLARIZABLE INTERACTION TRANSFORMER (EPIT)

We adopt PAINN (Schütt et al., 2021) as the backbone of our model, owing to its promising performance for molecular representation learning. Basically, our model takes as input the node features $\boldsymbol{H}_p^{(0)} = \boldsymbol{H}_p$, coordinates $\vec{\boldsymbol{X}}_p^{(0)} = \vec{\boldsymbol{X}}_p$ and all-zero vectors $\vec{\boldsymbol{V}}_p^{(0)} = \boldsymbol{0} \in \mathbb{R}^{N_p \times H \times 3}$, and outputs SE(3)-invariant features $\boldsymbol{H}_p^{(L)} \in \mathbb{R}^{N_p \times H}$ as well as SO(3)-equivariant vectors $\vec{\boldsymbol{V}}_p^{(L)} \in \mathbb{R}^{N_i \times H \times 3}$ after $L$-layer message passing ($p \in \{1, 2\}$).

Upon PAINN, we further introduce a **G**raph **T**ransformer **L**ayer (GTL) into each message passing block, to enhance the ability of detecting protein pockets. Given a pair of adjacent nodes $i, j$ with corresponding node features $\boldsymbol{h}_i, \boldsymbol{h}_j$ and edge feature $\boldsymbol{e}_{j \to i}$, we obtain a message from $j$ to $i$ through a fully-connected layer:

$$\boldsymbol{m}_{j \to i} = \mathrm{FC}(\boldsymbol{h}_j, \boldsymbol{e}_{j \to i}) \in \mathbb{R}^H. \tag{1}$$

Then we construct the graph transformer layer as follows:

$$\boldsymbol{q}_{j \to i}^{(u)}, \boldsymbol{k}_{j \to i}^{(u)}, \boldsymbol{v}_{j \to i}^{(u)} = \mathrm{FC}(\boldsymbol{m}_{j \to i}) \in \mathbb{R}^H, \ j \in \mathcal{N}_i, \ 1 \le u \le U, \tag{2}$$

$$\alpha_{j \to i}^{(u)} = \mathrm{softmax}_j \big( \frac{1}{\sqrt{H}} \langle \boldsymbol{q}_{j \to i}^{(u)}, \boldsymbol{k}_{j \to i}^{(u)} \rangle \big) \in \mathbb{R}, \tag{3}$$

$$\boldsymbol{o}_{j \to i}^{(u)} = \alpha_{j \to i}^{(u)} \boldsymbol{v}_{j \to i}^{(u)} \in \mathbb{R}^H, \tag{4}$$

$$\boldsymbol{m}'_{j \to i} = \mathrm{FC}(\mathrm{concat}_u(\boldsymbol{o}_{j \to i}^{(u)})) \in \mathbb{R}^H, \tag{5}$$

$$\boldsymbol{m}'_i = \frac{1}{|\mathcal{N}_i|} \sum_{j \in \mathcal{N}_i} \boldsymbol{m}'_{j \to i} \in \mathbb{R}^H. \tag{6}$$

Here, $U$ denotes the number of attention heads, $\mathrm{FC}(\cdot) = \mathrm{Linear} \circ \mathrm{SiLU} \circ \mathrm{Linear}(\cdot)$ denotes a fully-connected layer, $\langle \cdot, \cdot \rangle$ denotes the inner product operation, $\mathcal{N}_i$ collects the neighbors of node $i$, and $\mathrm{concat}_u(\boldsymbol{o}_{j \to i}^{(u)})$ indicates the concatenation of $\boldsymbol{o}_{j \to i}^{(u)}$ in terms of all values of $u$. The aggregated message $\boldsymbol{m}'_i$ will be used to update $\boldsymbol{H}$ and $\vec{\boldsymbol{V}}$ following the pipeline in PAINN. The intra-part architecture of EPIT is illustrated in Appendix.

So far, all the message passing processes are conducted only through intra-protein edges, without modeling the interaction between the receptor and the ligand. As they are in unbound states, it makes

no sense to construct inter-protein edges based on 3D coordinates similar to those intra-protein edges. Instead, we propose dense edge connections between every node pair of the receptor and the ligand, and only allow the invariant node features $\boldsymbol{H}_p^{(l)}$ (not $\vec{V}_p^{(l)}$) to be passed along the intra-protein edges. By defining $\boldsymbol{W}^{(l)} \in \mathbb{R}^{H \times H}$ as a learnable matrix of layer $l$, we derive the following updates:

$$\boldsymbol{\beta}_p^{(l)} = \text{sigmoid}(\mu(\boldsymbol{H}_p^{(l)} \boldsymbol{W}^{(l)} (\boldsymbol{H}_{\neg p}^{(l)})^{\top})) \in (0,1)^{N_p}, \tag{7}$$

$$\boldsymbol{H}_p^{'(l)} = \boldsymbol{H}_p^{(l)} + \boldsymbol{\beta}_p^{(l)} \odot \text{FC}(\boldsymbol{H}_p^{(l)}) \in \mathbb{R}^{N_p \times H}. \tag{8}$$

where $\neg p$ denotes the protein other than $p$, $\mu(\cdot)$ computes the average along the second dimension of the input matrix, $\odot$ implies element-wise multiplication, and $\text{FC}(\boldsymbol{H}_p^{(l)})$ is conducted on each row of $\boldsymbol{H}_p^{(l)}$ individually. The updated features $\boldsymbol{H}_p^{'(l)}$ are considered as the input of the next layer.

We have the following conclusion.

**Proposition 1** *The proposed model* EPIT *is independent SE(3)-equivariant and translation-invariant, indicating that if* $\vec{V}_1^{(L)}, \boldsymbol{H}_1^{(L)}, \vec{V}_2^{(L)}, \boldsymbol{H}_2^{(L)} = \text{EPIT}(\vec{X}_1, \boldsymbol{H}_1, \mathcal{E}_1, \vec{X}_2, \boldsymbol{H}_2, \mathcal{E}_2)$, *then, we have* $\vec{V}_1^{(L)} \boldsymbol{Q}_1, \boldsymbol{H}_1^{(L)}, \vec{V}_2^{(L)} \boldsymbol{Q}_2, \boldsymbol{H}_2^{(L)} = \text{EPIT}(\vec{X}_1 \boldsymbol{Q}_1 + \vec{t}_1, \boldsymbol{H}_1, \mathcal{E}_1, \vec{X}_2 \boldsymbol{Q}_2 + \vec{t}_2, \boldsymbol{H}_2, \mathcal{E}_2), \forall \boldsymbol{Q}_1, \boldsymbol{Q}_2 \in \text{SO}(3), \vec{t}_1, \vec{t}_2 \in \mathbb{R}^3$.

The proof is given in Appendix. Such property is desirable and it will ensure our later docking process to be generalizable to arbitrary poses of the input proteins.

### 3.3 Elliptic Paraboloid

We briefly discuss the rationale for choosing an elliptical paraboloid as the interface. On the one hand, using planes, spheres or other symmetric 2D manifolds to model the interface will easily cause the docking ambiguity issue since there are usually different ways to transform a 2D manifold from one pose to another. On the other hand, using complex manifolds may introduce bias and higher computational complexity to obtain the docking solutions. Thus we choose elliptic paraboloids to model the binding interface, which are appropriately defined in 3D space and their SE(3) transformations can be devised in a closed form. We provide mathematical details below.

For each point $\vec{x} = [x_1, x_2, x_3] \in \mathbb{R}^3$ located on the surface of a elliptic paraboloid, we have the standard form (without loss of generality, we might assume that it is symmetric about the $z$ axis):

$$ax_1^2 + bx_2^2 + cx_3 = 0, a, b \in \mathbb{R}_+, c \in \mathbb{R}. \tag{9}$$

The general form obtained by SE(3) transformations from the standard form belongs to a family of quadric surfaces:

$$\langle \boldsymbol{A}\vec{x}, \vec{x} \rangle + \langle \boldsymbol{b}, \vec{x} \rangle + c = 0, \ \boldsymbol{A} \in \mathbb{R}^{3 \times 3}, \boldsymbol{b} \in \mathbb{R}^3, c \in \mathbb{R}. \tag{10}$$

Actually, any SE(3) transformation of an elliptical paraboloid results in the change of the coefficients given by the following proposition.

**Proposition 2** *If we transform the elliptical paraboloid in Eq. 10 as* $\vec{x}' = \boldsymbol{Q}\vec{x} + \vec{t}$ *via a rotation matrix* $\boldsymbol{Q} \in \text{SO}(3)$ *and a translation vector* $\vec{t} \in \mathbb{R}^3$, *then the elliptical paraboloid becomes* $\langle \boldsymbol{A}'\vec{x}', \vec{x}' \rangle + \langle \boldsymbol{b}', \vec{x}' \rangle + c' = 0$, *where* $\boldsymbol{A}' = \boldsymbol{Q}\boldsymbol{A}\boldsymbol{Q}^{\top}, \boldsymbol{b}' = \boldsymbol{Q}\boldsymbol{b} - \boldsymbol{Q}(\boldsymbol{A} + \boldsymbol{A}^{\top})\boldsymbol{Q}^{\top}\vec{t}, c' = c + \vec{t}^{\top}\boldsymbol{Q}\boldsymbol{A}\boldsymbol{Q}^{\top}\vec{t} - \vec{t}^{\top}\boldsymbol{Q}\boldsymbol{b}$.

To fulfil our task, we construct a function $\Phi$ to predict an elliptic paraboloid of the general form for each of the input proteins:

$$\boldsymbol{A}_1, \boldsymbol{b}_1, c_1, \boldsymbol{A}_2, \boldsymbol{b}_2, c_2 = \Phi(\vec{X}_1, \boldsymbol{H}_1, \mathcal{E}_1, \vec{X}_2, \boldsymbol{H}_2, \mathcal{E}_2). \tag{11}$$

We further assume that $(\boldsymbol{A}_1, \boldsymbol{b}_1, c_1)$ shares the same standard form as $(\boldsymbol{A}_2, \boldsymbol{b}_2, c_2)$ since the ligand and the receptor do share the same interface after docking.

Specifically, the function $\Phi$ should require hard constraints to achieve the following property:

**Independent equivariance** For simplicity, we only take the ligand for example while the conclusion holds similarly for the receptor. For any SE(3) transformation of the input ligand, namely, $\forall \boldsymbol{Q} \in \text{SO}(3), \vec{t} \in \mathbb{R}^3$, we obtain the new predicted elliptic paraboloids by $\boldsymbol{A}_1', \boldsymbol{b}_1', c_1', \boldsymbol{A}_2, \boldsymbol{b}_2, c_2 = \Phi(\vec{V}_1 \boldsymbol{Q} + \vec{t}, \boldsymbol{H}_1, \mathcal{E}_1, \vec{V}_2, \boldsymbol{H}_2, \mathcal{E}_2)$. It should be guaranteed that $(\boldsymbol{A}_1', \boldsymbol{b}_1', c_1')$ is transformed from $(\boldsymbol{A}_1, \boldsymbol{b}_1, c_1)$ in the same way as stated in Proposition 2.

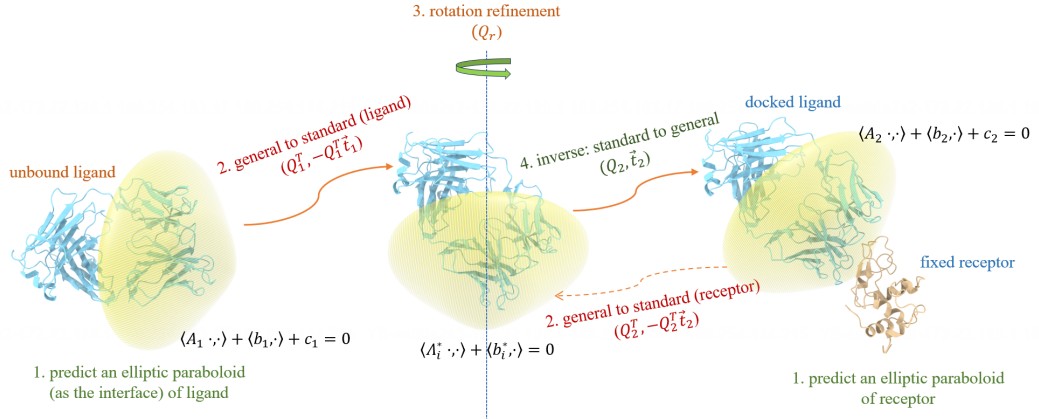

Figure 2: **Docking process via interface fitting.** We decompose the docking process into four steps: 1) predict elliptic paraboloid interfaces for two proteins respectively, 2) conduct general to standard transformations based on the ligand interface, 3) rotation refinement, 4) conduct standard to general transformations based on the receptor interface.

## 3.4 INTERFACE PREDICTION AND DOCKING VIA EPIT

In this subsection, we introduce how to implement the function $\Phi$ through our EPIT. The whole process of docking via interface fitting is illustrated in Figure 2. Based on Proposition 2, we can decompose the prediction of an elliptic paraboloid interface by first regressing its standard coefficients and then estimating the SE(3) transformation from the standard from.

We first obtain equivariant vectors $\vec{V}_1^{(L)} \in \mathbb{R}^{N_1 \times H \times 3}, \vec{V}_2^{(L)} \in \mathbb{R}^{N_2 \times H \times 3}$ and invariant features $\boldsymbol{H}_1^{(L)} \in \mathbb{R}^{N_1 \times H}, \boldsymbol{H}_2^{(L)} \in \mathbb{R}^{N_2 \times H}$ from the output of EPIT, then we aggregate the information along all nodes of the receptor (ligand) to get graph-level outputs:

$$\boldsymbol{F}_p = \sum_{j=1}^{N_p} \text{FC}(\boldsymbol{H}_p^{(L)}[j] \odot \text{sigmoid}(\mu(\boldsymbol{H}_p^{(L)}[j]\boldsymbol{W}'(\boldsymbol{H}_{\neg p}^{L})^{\top}))) \in \mathbb{R}^4, \ \boldsymbol{W}' \in \mathbb{R}^{H \times H}, p \in \{1, 2\},$$
(12)

$$\vec{\boldsymbol{E}}_p = \sum_{j=1}^{N_p} \text{LinearNoBias}(\boldsymbol{H}_p^{(L)}[j] \odot \vec{V}_p^{(L)}[j]) \in \mathbb{R}^{(3M+1) \times 3}, \ M \in \mathbb{N}_+, p \in \{1, 2\},$$
(13)

where $M$ (in our experiment, $M = 3$) is a hyperparameter, $\boldsymbol{H}_p^{(L)}[j]$ and $\vec{V}_p^{(L)}[j]$ denote the $j$-th rows of $\boldsymbol{H}_p^{(L)}$ and $\vec{V}_p^{(L)}$, respectively. It is proved that $\boldsymbol{F}_p$ still satisfies SE(3)-invariance and $\vec{\boldsymbol{E}}_p$ satisfies SO(3)-equivariance.

**Predicting Standard Forms** The standard coefficients are obtained with the invariant outputs $\boldsymbol{F}_1$ and $\boldsymbol{F}_2$ by:

$$\boldsymbol{\Lambda}_1^* = \boldsymbol{\Lambda}_2^* = \begin{bmatrix} \zeta(\boldsymbol{F}_1[0] + \boldsymbol{F}_2[0]) & & \\ & \zeta(\boldsymbol{F}_1[1] + \boldsymbol{F}_2[1]) & \\ & & 0 \end{bmatrix}, \boldsymbol{b}_1^* = \boldsymbol{b}_2^* = \begin{bmatrix} 0 \\ 0 \\ \boldsymbol{F}_1[2] + \boldsymbol{F}_2[2] \end{bmatrix}. \quad (14)$$

where $\zeta(\cdot)$ denotes the softplus operation. The standard form of the predicted elliptic paraboloid is given by:

$$\langle \boldsymbol{\Lambda}_p^* \vec{x}, \vec{x} \rangle + \langle \boldsymbol{b}_p^*, \vec{x} \rangle = 0, \ p \in \{1, 2\}. \quad (15)$$

**Predicting SE(3) Transformations** We first use the equivariant output $\vec{\boldsymbol{E}}_p$ to generate intermediate variables $\boldsymbol{R}_p, \vec{t}_p', p \in \{1, 2\}$ that maintain SO(3)-equivariance:

$$\boldsymbol{R}_p = \sum_{j=0}^{M-1} \vec{\boldsymbol{E}}_p[3j : 3(j+1)] \in \mathbb{R}^{3 \times 3}, \vec{t}_p = \vec{\boldsymbol{E}}_p[3M] \in \mathbb{R}^3, \ p \in \{1, 2\}. \quad (16)$$

Here $M$ aims at ensuring $\boldsymbol{R}_i$ to be non-singular. We should make sure $\det(\boldsymbol{R}_p) > 0$ to derive a rotation matrix instead of a reflection matrix, otherwise let $\boldsymbol{R}_p = -\boldsymbol{R}_p$. We then generate a $\mathrm{SO}(3)$-equivariant rotation matrix $\boldsymbol{Q}_p$ based on Proposition 3:

**Proposition 3** *Given a* $\mathrm{SO}(3)$-*equivariant matrix* $\boldsymbol{R} \in \mathbb{R}^{3\times3}$ *which satisfies* $\det(\boldsymbol{R}) > 0$*, we can generate a* $\mathrm{SO}(3)$-*equivariant rotation matrix* $\boldsymbol{Q}$ *from*

$$\boldsymbol{U} = (\boldsymbol{R}\boldsymbol{R}^{\top})^{1/2}, \boldsymbol{Q} = \boldsymbol{U}^{-1}\boldsymbol{R}.$$

As $\vec{t}_1, \vec{t}_2$ only satisfy $\mathrm{SO}(3)$-equivariance, we generate $\mathrm{SE}(3)$-equivariant vectors by simply adding the coordinate center $\vec{c}_l$ of the ligand and $\vec{c}_r$ of the receptor:

$$\vec{t}_1 = \vec{t}_1 + \vec{c}_l \in \mathbb{R}^3, \ \vec{t}_2 = \vec{t}_2 + \vec{c}_r \in \mathbb{R}^3. \tag{17}$$

Now we obtain the transformations $(\boldsymbol{Q}_1, \vec{t}_1)$ for the ligand, and $(\boldsymbol{Q}_2, \vec{t}_2)$ for the receptor, and their corresponding elliptic paraboloids of the general form based on Proposition 2:

$$\boldsymbol{A}_p = \boldsymbol{Q}_p\boldsymbol{\Lambda}_p^*\boldsymbol{Q}_p^{\top}, \ \boldsymbol{b}_p = \boldsymbol{Q}_p\boldsymbol{b}_p^* - 2\boldsymbol{Q}_p\boldsymbol{\Lambda}_p\boldsymbol{Q}_p^{\top}\vec{t}_p, \ c_p = \vec{t}_p^{\top}\boldsymbol{Q}_p\boldsymbol{\Lambda}_p^*\boldsymbol{Q}_p^{\top}\vec{t}_p - \vec{t}_p^{\top}\boldsymbol{Q}_p\boldsymbol{b}_p^*, \ p \in \{1, 2\}. \tag{18}$$

In our task formulation, we fix the receptor and only dock the ligand towards the receptor. Since $(\boldsymbol{Q}_1, \vec{t}_1)$ and $(\boldsymbol{Q}_2, \vec{t}_2)$ are the transformations from the general interface to the standard interface, we re-formulate their values as the relative transformation between the ligand and the receptor based on Proposition 4:

**Proposition 4** *Denote* $\boldsymbol{Q}_{1\to2} \in \mathrm{SO}(3), \vec{t}_{1\to2} \in \mathbb{R}^3$ *as the relative transformation of the unbound ligand to the docked pose with the receptor fixed, then we have*

$$\boldsymbol{Q}_{1\to2} = \boldsymbol{Q}_2\boldsymbol{Q}_1^{\top}, \ \vec{t}_{1\to2} = \vec{t}_2 - \boldsymbol{Q}_2\boldsymbol{Q}_1^{\top}\vec{t}_1.$$

### 3.5 ROTATION REFINEMENT

According to Section 3.3, we only restrict quadratic coefficients of the elliptic paraboloid with an inequality such that nodes should be inside (or outside) the interface utmostly, which is considered to be a loose constraint and leads to unexpected degrees of freedom in x-y directions, after the interface of the ligand being transformed to the standard form. So we need an additional rotation to refine x-y angles. Specifically, denote $\theta = \boldsymbol{F}_1[3] - \boldsymbol{F}_2[3] \in \mathbb{R}$, we compute the rotation matrix for refinement:

$$\boldsymbol{Q}_r = \begin{bmatrix} \cos(\theta) & \sin(\theta) & 0 \\ -\sin(\theta) & \cos(\theta) & 0 \\ 0 & 0 & 1 \end{bmatrix} \in \mathrm{SO}(3). \tag{19}$$

Then the final docking transformation with rotation refinement conducted is given by:

$$\boldsymbol{Q}_{1\to2} = \boldsymbol{Q}_2\boldsymbol{Q}_r\boldsymbol{Q}_1^{\top} \in \mathrm{SO}(3), \ \vec{t}_{1\to2} = \vec{t}_2 - \boldsymbol{Q}_2\boldsymbol{Q}_r\boldsymbol{Q}_1^{\top}\vec{t}_1 \in \mathbb{R}^3. \tag{20}$$

### 3.6 TRAINING OBJECTIVE

**Fitness Loss**. Suppose $\vec{X}_1^*, \vec{X}_2^*$ as the coordinates in the docked state, from which we select all pairs of nodes $\vec{Y}_1^*, \vec{Y}_2^* \in \mathbb{R}^{K\times3}$ such that $||\vec{Y}_{1k}^* - \vec{Y}_{2k}^*|| < 8\mathring{A}, \ \forall 1 \leq k \leq K$, where $K$ denotes the number of such point pairs. Then we regard the midpoints of those pairs to be functional sites, $\vec{P}_1^* = \vec{P}_2^* = \frac{1}{2}(\vec{Y}_1^* + \vec{Y}_2^*)$. In the unbound state, those point sets should follow transformations of corresponding proteins, we denote them as $\vec{P}_1$ and $\vec{P}_2$. For clarity, if $\vec{X}_1 = \vec{X}_1^*\boldsymbol{Q} + \vec{t}$, then $\vec{P}_1 = \vec{P}_1^*\boldsymbol{Q} + \vec{t}$. Naturally, we expect those pockets to be exactly located on our predicted interface. Furthermore, we propose a stricter constraint to represent how interface acts with each other, that is, we choose to minimize the distance between pockets and the tangent plane at the peak of the interface. Formally, denote $\vec{r}_p \in \mathbb{R}^3$ as the peak of the predicted interface and $\vec{n}_p \in \mathbb{R}^3$ as the normal vector of the tangent plane at the peak ($p \in \{1, 2\}$), we enforce the optimization objective as follows:

$$\mathcal{L}_{\mathrm{fit}} = \frac{1}{K} \sum_{p\in\{1,2\}} \sum_{k=1}^{K} (||\langle\boldsymbol{A}_p\vec{P}_p[k], \vec{P}_p[k]\rangle + \langle\boldsymbol{b}_p, \vec{P}_p[k]\rangle + c_p||^2 + |\langle\vec{P}_p[k] - \vec{t}_p, \vec{n}_p\rangle|). \tag{21}$$

**Overlap Loss**. One of the crucial issues in previous works is that they encounter steric clashes during the docking process. We address this issue by restricting receptor and ligand to either side of the interface in the standard form, such that the interface spatially isolates protein pairs. Denote $\phi_p(\cdot) = \langle \boldsymbol{A}_p\cdot, \cdot \rangle + \langle \boldsymbol{b}_p, \cdot \rangle + c_p$, $p \in \{1, 2\}$, we can tell whether a node $\vec{\boldsymbol{x}}$ is inside or outside of the interface $p$ by checking the sign of the formula $\phi_p(\vec{\boldsymbol{x}})$, from which we define the training objective:

$$\mathcal{L}_{\text{overlap}} = \min\{\frac{1}{N_1}\sum_{j=1}^{N_1}(\sqrt{\text{ReLU}(\phi_1(\vec{\boldsymbol{X}}_1[j]))} + \frac{1}{N_2}\sum_{j=1}^{N_2}(\sqrt{\text{ReLU}(-\phi_2(\vec{\boldsymbol{X}}_2[j]))},$$
$$\frac{1}{N_1}\sum_{j=1}^{N_1}(\sqrt{\text{ReLU}(-\phi_1(\vec{\boldsymbol{X}}_1[j]))} + \frac{1}{N_2}\sum_{j=1}^{N_2}(\sqrt{\text{ReLU}(\phi_2(\vec{\boldsymbol{X}}_2[j]))}\}. \tag{22}$$

**Refinement Loss**. As described in Section 3.5, we conduct a rotation refinement when the interface is transformed to the standard form. We expect to minimize the distance between the docked pockets in the standard form (denoted as $\hat{\vec{\boldsymbol{P}}}_1$ and $\hat{\vec{\boldsymbol{P}}}_2$) only in x and y dimensions, which can be measured by Kabsch algorithm (Kabsch, 1976). For clarity, we have $\hat{\vec{\boldsymbol{P}}}_1[k] = \boldsymbol{Q}_1^\top \vec{\boldsymbol{P}}_1[k] - \boldsymbol{Q}_1^\top \vec{\boldsymbol{t}}_1$, $\hat{\vec{\boldsymbol{P}}}_2[k] = \boldsymbol{Q}_2^\top \vec{\boldsymbol{P}}_2[k] - \boldsymbol{Q}_2^\top \vec{\boldsymbol{t}}_2$ $(1 \le k \le K)$, and the refinement loss is defined as

$$\mathcal{L}_{\text{ref}} = ||\boldsymbol{Q}_r[:, 2, :2] - \text{Kabsch}(\hat{\vec{\boldsymbol{P}}}_1[:2], \hat{\vec{\boldsymbol{P}}}_2[:2])||^2. \tag{23}$$

**Dock Loss**. Since the degrees of freedom of rigid docking is limited, we avoid applying the MSE loss between predicted coordinates and the ground truth to prevent a large number of floating-point calculation errors. During the training process, as we randomly sample the rotation matrix $\boldsymbol{Q}_{gt} \in \text{SO}(3)$ and the translation vector $\vec{\boldsymbol{t}}_{gt} \in \mathbb{R}^3$ to get the unbound ligand pose $\vec{\boldsymbol{X}}_{1i} = \boldsymbol{Q}_{gt}\vec{\boldsymbol{X}}_{1i}^* + \vec{\boldsymbol{t}}_{gt}$, $1 \le i \le N_1$, it's possible to directly evaluate the transformation error by

$$\mathcal{L}_{\text{dock}} = ||\boldsymbol{Q}_{1\to2} - \boldsymbol{Q}_{gt}^\top||^2 + ||\vec{\boldsymbol{t}}_{1\to2} + \boldsymbol{Q}_{gt}^\top\vec{\boldsymbol{t}}_{gt}||^2. \tag{24}$$

## 4 EXPERIMENTS

### 4.1 EXPERIMENTAL SETUP

**Baselines** We mainly compare our model ElliDock with one of the-state-of-the-art protein complex structure prediction model Alphafold-Multimer[1] (Evans et al., 2021), the template-based docking server HDock[2] (Yan et al., 2020), the regression-based docking model EquiDock[3] (Ganea et al., 2021) and the diffusion-based docking model DiffDock-PP[4] (Ketata et al., 2023). Note that, we train EquiDock and DiffDock-PP from scratch with their recommended hyperparameters before testing on the corresponding dataset, while we use pre-trained models of other baselines.

**Datasets** We utilize the following two datasets for our experiments:

**Docking benchmark version 5.** We first leverage the Docking benchmark version 5 database (DB5.5) (Vreven et al., 2015), which is a gold standard dataset with 253 high-quality complex structures. We use the data split provided by EquiDock, with partitioned sizes 203/25/25 for train/validation/test.

**The Structural Antibody Database.** To further simulate real drug design scenarios and study the performance of our method on heterodimers, we select the Structural Antibody Database (SAbDab) (Dunbar et al., 2014) for training. SAbDab is a database of thousands of antibody-antigen complex structures that updates on a weekly basis. Datasets are split based on sequence similarity measured by MMseqs2 (Steinegger & Söding, 2017), with partitioned sizes 1,781/300/0 for train/validation/test. We then use the RAbD database (Adolf-Bryfogle et al., 2018), a curated benchmark on antibody design with 54 complexes (note that PDB 5d96 will result in NaN IRMSD, so it has been removed here), as the test set of antibody-antigen complexes. To prevent data leakage, we put proteins with sequence correlation greater than 0.8 into the same cluster and ensure that no clusters included in RAbD appear in the training set and the validation set.

---

[1]https://github.com/YoshitakaMo/localcolabfold.
[2]http://huanglab.phys.hust.edu.cn/software/hdocklite.
[3]https://github.com/octavian-ganea/equidock_public.
[4]https://github.com/ketatam/DiffDock-PP.

Table 1: **Complex prediction results (DB5.5 test).** Note that * means we use the pre-trained model, otherwise we train from scratch on the corresponding dataset before testing.

| Methods | CRMSD (↓) | | | IRMSD (↓) | | | DockQ (↑) | | | Inference time |
|---|---|---|---|---|---|---|---|---|---|---|
| | median | mean | std | median | mean | std | median | mean | std | |
| HDock* | **0.327** | **3.745** | 7.139 | **0.289** | **3.548** | 6.842 | **0.981** | **0.791** | 0.386 | 11478.4 |
| Multimer* | 1.987 | 7.081 | 7.258 | 1.759 | 7.141 | 7.889 | 0.629 | 0.482 | 0.418 | 56762.5 |
| EquiDock | 14.136 | 14.726 | 5.312 | 11.971 | 13.233 | 4.931 | 0.036 | 0.044 | 0.034 | 60.1 |
| DiffDock-PP | 14.109 | 15.419 | 8.160 | 15.060 | 16.881 | 11.397 | 0.025 | 0.035 | 0.033 | 2103.1 |
| **ElliDock** | 12.995 | 14.413 | 6.780 | 11.134 | 12.480 | 4.966 | 0.037 | 0.060 | 0.060 | **36.7** |

Table 2: **Complex prediction results (SAbDab test).** Note that * means we use the pre-trained model, otherwise we train from scratch on the corresponding dataset before testing.

| Methods | CRMSD (↓) | | | IRMSD (↓) | | | DockQ (↑) | | | Inference time |
|---|---|---|---|---|---|---|---|---|---|---|
| | median | mean | std | median | mean | std | median | mean | std | |
| HDock* | **0.323** | **2.792** | 6.798 | **0.262** | **2.677** | 6.803 | **0.982** | **0.861** | 0.310 | 37328.8 |
| Multimer* | 13.598 | 14.071 | 6.091 | 12.969 | 12.548 | 5.435 | 0.050 | 0.104 | 0.172 | 197503.1 |
| EquiDock | 14.301 | 15.032 | 5.548 | 12.700 | 12.712 | 5.390 | 0.034 | 0.055 | 0.067 | 274.5 |
| DiffDock-PP | 11.764 | 12.560 | 6.241 | 12.207 | 12.401 | 6.353 | 0.045 | 0.076 | 0.090 | 8308.7 |
| **ElliDock** | 11.541 | 13.402 | 6.306 | 11.319 | 11.550 | 4.681 | 0.054 | 0.082 | 0.084 | **91.2** |

**Evaluation metrics** We apply Complex Root Mean Squared Deviation (CRMSD), Interface Root Mean Squared Deviation (IRMSD) and DockQ (Basu & Wallner, 2016) to evaluate the predicted complex quality. See Appendix for specific definitions.

## 4.2 COMPARISONS OF THE INFERENCE TIME

Given real world applications (for example, simulating the binding process of artificially edited proteins), faster inference time will nicely support a large number of simulated docking processes within an acceptable time, such that we can locate the desirable protein candidate from its huge amino acid space. Here we conduct the inference time test for different methods on two test sets and results are shown in Table 1, 2, accordingly. For fairness, all models are tested under the same environment on one single GPU (instead of HDock, which can only run on CPU).

As expected, the traditional template-based docking method HDock takes an extremely long time to run because it needs to generate tens of thousands of structure templates for further update. Multimer is a deep learning model, yet it requires multiple sequence alignment during inference with GB level parameters, which may attribute to its longest inference time. In comparison, generative-based and regression-based deep learning models are 10~1,000 times faster. Among them, DiffDock-PP runs slower because it requires dozens of diffusion steps, each step requires re-running the model. On the contrary, ElliDock and EquiDock are both regression models without an order of magnitude difference in speed. We show ElliDock is 2~3 times faster than EquiDock, which is more likely due to the amount of model parameters, number of model layers and so forth.

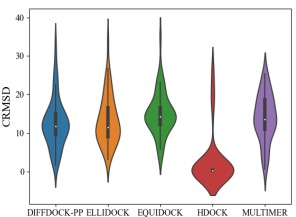

Figure 3: CRMSD of antibody-antigen docking on SAbDab test set.

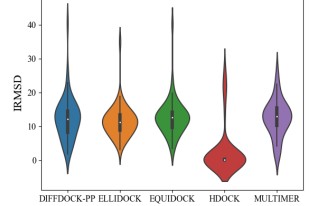

Figure 4: IRMSD of antibody-antigen docking on SAbDab test set.

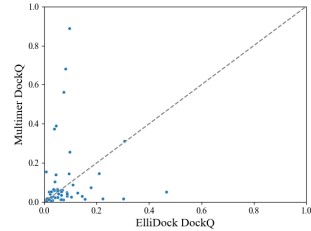

Figure 5: The comparison of DockQ between ElliDock and Alphafold-Multimer on SAbDab test set.

### 4.3 RESULTS OF THE COMPLEX PREDICTION

The complex prediction results on the DB5.5 test set and RAbD are illustrated in Table 1 and Table 2 respectively, with statistically processed data shown in Figure 3, 4, 5. We show that, ElliDock outperforms the other regression-based model, EquiDock, on both DB5.5 and SAbDab based on IRMSD and CRMSD. Meanwhile, Our method beats DiffDock-PP on all metrics on DB5.5, while the CRMSD metric is slightly inferior to the latter on SAbDab. We assume that this is because the training objective of DiffDock-PP is completely based on the entire molecules, on the contrary, our method focuses more on pockets in localized regions of the protein. For the comparison with Alphafold-Multimer, we illustrate in Figure 5 that ElliDock has better prediction results on most samples on SAbDab, yet Multimer achieves extremely amazing accuracy on specific samples, which may be attributed to the powerful representation ability of Alphafold. For pre-trained models like HDock, we respect for its promising performance, yet we have some concerns about its possible data leakage.

### 4.4 ABLATION STUDIES

**Training objectives.** To illustrate contributions of different training objectives, we conduct multiple experiments and rule out some of the training objectives each time. Models are trained and tested on DB5.5 only.

**Without rotation refinement.** With a view to verify the validity of the rotation refinement, we removed $Q_r$ in the calculation of the docking transformation $Q(\vec{X}_1|\vec{X}_2), \vec{t}(\vec{X}_1|\vec{X}_2)$. Similarly, we train and test the model on DB5.5.

Ablation study results are shown in Table 3. We conclude that $\mathcal{L}_{\text{fit}}$ and the rotation refinement influence the docking process by searching for better binding interface, with a great impact on IRMSD. Another training objective, $\mathcal{L}_{\text{overlap}}$,
designed for tackling conformation overlap, does help to find proper docking positions for ligand, which results in lower CRMSD.

Table 3: **Ablation studies.** We report CRMSD median and IRMSD median of the corresponding best validated model on DB5.5.

| Methods | CRMSD ($\downarrow$) | IRMSD ($\downarrow$) |
|---|---|---|
| Full model | **13.00** | **11.13** |
| - $\mathcal{L}_{\text{fit}}$ | 13.20 | 13.02 |
| - $\mathcal{L}_{\text{overlap}}$ | 13.95 | 12.35 |
| - $\mathcal{L}_{\text{ref}}$ | 13.23 | 12.49 |
| - $\mathcal{L}_{\text{fit}}, \mathcal{L}_{\text{overlap}}$ | 13.98 | 13.24 |
| - $\mathcal{L}_{\text{fit}}, \mathcal{L}_{\text{ref}}$ | 14.71 | 13.94 |
| - $\mathcal{L}_{\text{overlap}}, \mathcal{L}_{\text{ref}}$ | 14.48 | 11.71 |
| - $Q_r$ | 13.05 | 13.22 |

## 5 CONCLUSION

We propose a new method for rigid protein-protein docking based on elliptic paraboloid interfaces prediction and fitting, which is strongly competitive with state-of-the-art learning-based methods for particularly antibody-antigen docking. Meanwhile, we design a global level, pairwise SE(3)-equivariant message passing network called EPIT, serving as a solution to message passing and updating through nodes when the graph cannot be efficiently constructed as connected.

In the future, we look forward to incorporate more domain knowledge to our method, for example, chemical properties on the protein surface. We also hope the interface-based docking paradigm can give an insight into rigid docking or further flexible docking, and provide more possible research directions.

**Limitations.** First, our model is targeted on rigid protein-protein docking regardless of conformational changes of receptor and ligand, which may not be applicable in all scenarios. A potentially more valuable follow-up direction is to study flexible docking, which takes the conformation changes during docking process into consideration. Moreover, we still use soft constraints in our model, so that the prediction results may not fully meet all properties that we require. Hard constraints can be further implemented in the model architecture for improvement.

ACKNOWLEDGMENTS

This work is supported by the National Science and Technology Major Project under Grant 2020AAA0107300, the National Natural Science Foundation of China (No. 61925601, 62376276), and Beijing Nova Program (20230484278).

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

## APPENDIX

## A    SOURCE CODE

Our source code is available at `https://github.com/yaledeus/ElliDock`.

## B    DETAILS OF PROTEIN REPRESENTATION

We construct the protein graph at residue level, each node in the graph corresponds to a residue that makes up the protein and each edge between two nodes represents an interaction. Here we show details on how to extract node features and edge features from the raw protein data.

### B.1    NODE FEATURES

**Residue Type Embedding**. The key to different amino acid types lies in their side chains, which plays an important role in biological functions of the protein. Since most natural proteins are composed of only 20 amino acids, we can establish an embedding of residue types to $D$ dimension vectors to capture side chain features.

**Relative Positional Embedding**. A motif is a fundamental structure representing specific biological functions with a group of adjacent residues. So the relative position of residues can reflect protein functionality. Given a protein chain composed of $N$ residues, we construct relative positional embedding for residue $i$ ($i \in \{0, 1, \cdots, N-1\}$) as follows:

$$\mathbf{RP}_i = \text{concat}_d([i\sin(10000^{\frac{-2d}{D}}), i\cos(10000^{\frac{-2d}{D}})]) \in \mathbb{R}^D,\ d \in \{0, 1, \cdots, \frac{D}{2} - 1\}. \quad (25)$$

### B.2    EDGE FEATURES

We adopt protein representation methods in EquiDock to construct relative position, relative orientation and distance based edge features. Here we introduce a new $\text{SE}(3)$-invariant edge feature to better understand the protein structure-function relationship.

**N-order Resultant Force**. The structure of a protein is essentially due to the interactions between residues and atoms. Based on this, we establish the force relationship between residues. Denote $\vec{s}_{i \to j}$ as the vector starting at node $j$ and ending at node $i$, $\vec{n}_{i,\alpha} = \text{norm}(\sum_{j \in \mathcal{N}_i} ||\vec{s}_{i \to j}||^{-\alpha-1}\vec{s}_{i \to j})$ as the direction of the resultant $\alpha$-order force from neighbors on node $i$. Then we calculate the inner product between resultant forces of node $i$ and $j$ as the $\text{SE}(3)$-invariant edge feature

$$f_{i,\alpha} = \langle \vec{n}_{i,\alpha}, \vec{n}_{j,\alpha} \rangle \in \mathbb{R},\ \alpha \in \{2, 3, 4\}. \quad (26)$$

## C    MODEL ARCHITECTURES

We show the intra-part architecure of our model EPIT in Figure 6.

## D    PROOFS OF THE MAIN PROPOSITIONS

**Proof of Proposition 1**    In the work of PAINN (Schütt et al., 2021), it has been proved that if $\vec{V}, H' = \text{PAINN}(\vec{X}, H)$, we have $\vec{V}Q, H' = \text{PAINN}(\vec{X}Q + \vec{t}, H)$, $\forall Q \in \text{SO}(3), \vec{t} \in \mathbb{R}^3$, here we only need to prove that the graph transformer layer and the inter-part architecture have the same properties.

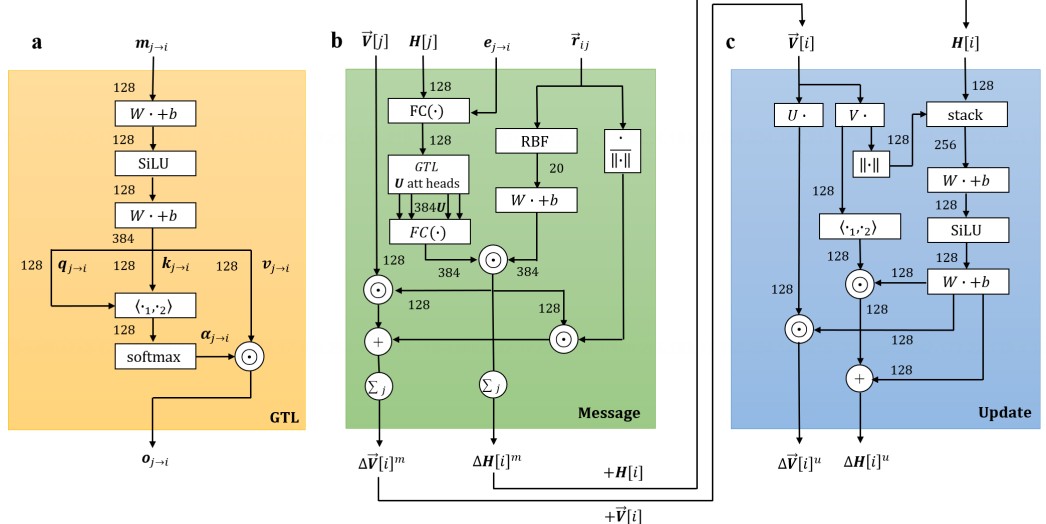

Figure 6: The **intra-part** architecture of EPIT with the Graph Transformer Layer (GTL) (a) as well as the message (b) and update blocks (c) of the equivariant message passing. The message block and the update block together form one EPIT layer.

First, we prove the graph transformer layer satisfies SE(3)-invariance. Given SE(3)-invariance feature $m_{j\to i}$, from Equation 2, we know that $q_{j\to i}^{(u)}, k_{j\to i}^{(u)}, v_{j\to i}^{(u)}$ still satisfy SE(3)-invariance since $m_{j\to i}$ remains the same for any roto-translation transformation. Thus GTL satisfies SE(3)-invariance, which means the intra-part architecture of EPIT satisfies SE(3)-invariance for features and SO(3)-equivariance for vectors as well.

Then we prove the inter-part architecture satisfies SE(3)-invariance. From Equation 7, we know that $\beta_p^{(l)}, H_p^{'(l)}$ satisfy SE(3)-invariance since $H_p^{(l)}$ is the output of the intra-part layer of EPIT, satisfying SE(3)-invariance. We have proved that the inter-part architecture of EPIT satisfies SE(3)-invariance.

**Proof of Proposition 2** From $\vec{x}' = Q\vec{x} + \vec{t}$ we have $\vec{x} = Q^\top(\vec{x}' - \vec{t})$. Substitute $\vec{x}$ into the surface equation before transformation:

$$(\vec{x}'^\top - \vec{t}^\top)QAQ^\top(\vec{x}' - \vec{t}) + (\vec{x}'^\top - \vec{t}^\top)Qb + c = 0. \tag{27}$$

$$\vec{x}'^\top QAQ^\top\vec{x}' + \vec{x}'^\top(Qb - Q(A + A^\top)Q^\top\vec{t}) + \vec{t}^\top QAQ^\top\vec{t} - \vec{t}^\top Qb + c = 0. \tag{28}$$

It illustrates that $\vec{x}'$ is on the elliptic paraboloid with parameters:

$$A' = QAQ^\top, b' = Qb - Q(A + A^\top)Q^\top\vec{t}, c' = c + \vec{t}^\top QAQ^\top\vec{t} - \vec{t}^\top Qb. \tag{29}$$

**Proof of Proposition 3** Denote $L = RR^\top$ (then $U = L^{1/2}$), we can easily show that $L$ is a positive semidefinite matrix.

$$\forall\vec{v} \in \mathbb{R}^3, \ \vec{v}^\top L\vec{v} = \vec{v}^\top RR^\top\vec{v} = (R^\top\vec{v})^\top(R^\top\vec{v}) \geq 0. \tag{30}$$

For clarity. Since $L$ is symmetric, denote $L = S\Lambda S^\top$ where $S$ is orthogonal and $\Lambda$ is diagonal, then $U = L^{1/2} = S\Lambda^{1/2}S^\top$ is defined as the matrix square root of $L$, which is unique since $L$ satisfies positive semidefinite.

For $\forall W \in \mathrm{SO}(3)$, denote $R' = WR, L' = R'R'^\top, U' = L'^{1/2}$, we have

$$U' = (R'R'^\top)^{1/2} = (WLW^\top)^{1/2} = (WS\Lambda S^\top W^\top)^{1/2} = (WS)\Lambda^{1/2}(WS)^\top = WUW^\top. \tag{31}$$

Then we can derive that $Q$ satisfies SO(3)-equivariance:

$$Q' = U'^{-1}R' = (WUW^\top)^{-1}WR = WU^{-1}W^\top WR = W(U^{-1}R) = WQ. \tag{32}$$

Next, we show that $\boldsymbol{Q}$ is orthogonal:

$$
\begin{aligned}
\boldsymbol{Q}^\top \boldsymbol{Q} &= (\boldsymbol{U}^{-1}\boldsymbol{R})^\top \boldsymbol{U}^{-1}\boldsymbol{R} \\
&= \boldsymbol{R}^\top (\boldsymbol{S}\Lambda^{1/2}\boldsymbol{S}^\top)^\top (\boldsymbol{S}\Lambda^{1/2}\boldsymbol{S}^\top)\boldsymbol{R} \\
&= \boldsymbol{R}^\top \boldsymbol{S}\Lambda^{-1}\boldsymbol{S}^\top \boldsymbol{R} \\
&= \boldsymbol{R}^\top \boldsymbol{L}^{-1}\boldsymbol{R} \\
&= \boldsymbol{R}^\top (\boldsymbol{R}\boldsymbol{R}^\top)^{-1}\boldsymbol{R} \\
&= \boldsymbol{I}.
\end{aligned}
\tag{33}
$$

Finally, since $\det(\boldsymbol{R}) > 0, \det(\boldsymbol{L}) = \det(\boldsymbol{R}\boldsymbol{R}^\top) = \det(\boldsymbol{R})^2 > 0$, we have

$$
\det(\boldsymbol{Q}) = \det(\boldsymbol{U}^{-1}\boldsymbol{R}) = \det(\boldsymbol{U}^{-1})\det(\boldsymbol{R}) = \det(\boldsymbol{L})^{-1/2}\det(\boldsymbol{R}) > 0.
\tag{34}
$$

So we assert $\det(\boldsymbol{Q}) = 1$, which means $\boldsymbol{Q}$ is a rotation matrix with $\mathrm{SO}(3)$-equivariance.

**Proof of Proposition 4**    Denote $I_1^*, I_2^*$ as the corresponding standard-form elliptic paraboloid surfaces of ligand and receptor, we can derive the transformation from general form to standard form:

$$
\boldsymbol{Q}(I_p|I_p^*) = \boldsymbol{Q}_p,\ \vec{\boldsymbol{t}}(I_p|I_p^*) = \vec{\boldsymbol{t}}_p,\ \boldsymbol{Q}(I_p^*|I_p) = \boldsymbol{Q}_p^\top,\ \vec{\boldsymbol{t}}(I_p^*|I_p) = -\boldsymbol{Q}_p^\top\vec{\boldsymbol{t}}_p,\ p \in \{1, 2\}.
\tag{35}
$$

When the receptor is fixed, the transformation of the ligand is the superposition of 1) the transformation from general to standard of its predicted elliptic paraboloid and 2) the transformation from standard to general of receptor's predicted elliptic paraboloid. We show that:

$$
\boldsymbol{Q}_{1\to 2} = \boldsymbol{Q}(I_2|I_2^*) \circ \boldsymbol{Q}(I_1^*|I_1) = \boldsymbol{Q}_2\boldsymbol{Q}_1^\top,
\tag{36}
$$

$$
\vec{\boldsymbol{t}}_{1\to 2} = \boldsymbol{Q}(I_2|I_2^*) \circ \vec{\boldsymbol{t}}(I_1^*|I_1) + \vec{\boldsymbol{t}}(I_2|I_2^*) = \vec{\boldsymbol{t}}_2 - \boldsymbol{Q}_2\boldsymbol{Q}_1^\top\vec{\boldsymbol{t}}_1.
\tag{37}
$$

# E    MORE EXPERIMENTAL DETAILS

**Training details**    Our models are trained and tested on an Intel(R) Xeon(R) Gold 5218 CPU @ 2.30GHz and a NVIDIA GeForce RTX 2080 Ti GPU. We save the top-10 models with the lowest loss evaluated on the whole validation set and apply early stopping with the patience of 8 epochs. After the training process, we test CRMSD and IRMSD on the test set of all 10 best validated models and record the best performance. During training process, we randomly select $\mathrm{SO}(3)$ rotation and $\mathbb{R}^3$ translation to apply to ligand coordinates and get the unbound structures as the input for the model.

**Metrics**    We give specific definitions of CRMSD, IRMSD and DockQ below.

**CRMSD.** Given the ground truth and predicted complex structure $\vec{\boldsymbol{X}}^*, \vec{\boldsymbol{X}} \in \mathbb{R}^{N\times 3}, N = N_1 + N_2$, we first conduct point cloud registration by Kabsch algorithm (Kabsch, 1976), and compute CRMSD $= \sqrt{\frac{1}{N}||\vec{\boldsymbol{X}}^* - \vec{\boldsymbol{X}}||^2}$.

**IRMSD.** Interface RMSD is similar to CRMSD but only computed among pocket points. We have already obtained the ground truth pockets $\vec{\boldsymbol{P}}_1^*, \vec{\boldsymbol{P}}_2^*$ for ligand and receptor, then we select nodes with the same index as $\vec{\boldsymbol{P}}_1^*, \vec{\boldsymbol{P}}_2^*$ from the predicted complex structure, denoted as $\vec{\boldsymbol{P}}_1', \vec{\boldsymbol{P}}_2'$. After concatenating $\vec{\boldsymbol{P}}_1^*, \vec{\boldsymbol{P}}_2^*$ and $\vec{\boldsymbol{P}}_1', \vec{\boldsymbol{P}}_2'$ respectively we get the pocket nodes of the complex, $\vec{\boldsymbol{P}}^*, \vec{\boldsymbol{P}}' \in \mathbb{R}^{2K\times 3}$. We compute IRMSD in the same way as CRMSD with ground truth and predicted pocket nodes $\vec{\boldsymbol{P}}^*$ and $\vec{\boldsymbol{P}}'$.

**DockQ.** DockQ is a weighted average of three terms including contact accuracy, interface RMSD and ligand RMSD, which is a quality measure for protein-protein docking models. We compute DockQ with the public tool[5].

---

[5]https://github.com/bjornwallner/DockQ.

**Test sets**   All test samples can be found in PDB (Burley et al., 2017), here we only list the entry IDs of complexes in the corresponding test set.

**DB5.5 test set.** 1AVX, 1H1V, 1HCF, 1IRA, 1JIW, 1JPS, 1MLC, 1N2C, 1NW9, 1QA9, 1VFB, 1ZHI, 2A1A, 2A9K, 2AYO, 2SIC, 2SNI, 2VDB, 3SZK, 4POU, 5C7X, 5HGG, 5JMO, 6B0S, 7CEI.

**SAbDab test set (RAbD).** 3MXW, 3O2D, 3K2U, 3BN9, 3HI6, 4DVR, 4G6J, 1FE8, 4ETQ, 4FFV, 1NCB, 3W9E, 3RKD, 5BV7, 5HI4, 5J13, 4CMH, 1N8Z, 3L95, 3S35, 1W72, 3NID, 4XNQ, 1A2Y, 4KI5, 1IC7, 4OT1, 1A14, 5D93, 5GGS, 1UJ3, 4DTG, 3CX5, 2ADF, 2B2X, 4G6M, 2XWT, 4FQJ, 2DD8, 5L6Y, 5MES, 2XQY, 2VXT, 4LVN, 5NUZ, 4H8W, 4YDK, 5B8C, 5EN2, 1OSP, 4QCI, 5F9O, 1IQD, 2YPV.

It should be noted that HDock is unable to conduct docking on PDB 5HI4 and PDB 5MES, thus the results of HDock reported in Table 2 have excluded the two samples.

**Hyperparameters**   We show hyperparameter choices for training in Table 4 and bold the best performing ones after evaluating on the corresponding validation set.

**Visualization**   The comparison of the predicted complex structures from different methods on PDB 1W72 has been illustrated in Figure 7. We also provide visualization of elliptic paraboloid interfaces predicted by ElliDock on protein complex samples in Figure 8.

Table 4: **Hyperparameters of the training process.** The bold marks represent the optimal parameters among their candidates based on validation results.

| Params | DB5.5 | SAbDab |
|---|---|---|
| learning rate | 2e-4 | 2e-4, **3e-4**, 5e-4 |
| layers of EPIT $L$ | **2**, 3 | 2, 3, 4, **5** |
| embed size $D$ | 64 | 64 |
| hidden size $H$ | 128 | 128 |
| attention heads $U$ | 16 | 4, 8, 16, **20** |
| neighbors | 10 | 10 |
| RBF size | 20 | 20 |
| radial cut ($\mathring{A}$) | 3.0 | 1.0, 2.0, **3.0** |
| dropout rate | 0.1 | 0.1 |

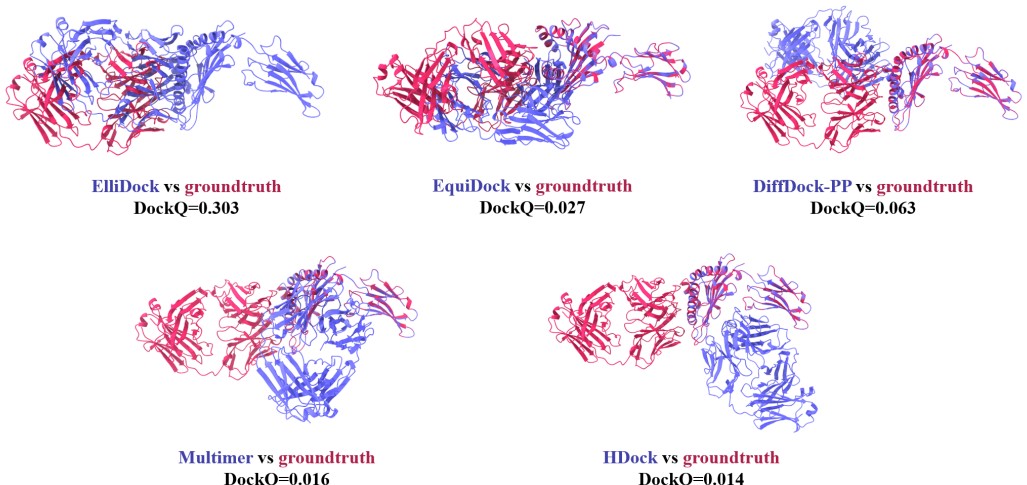

Figure 7: Visualization of the predicted complex structures from different methods on PDB 1W72 (from RAbD).

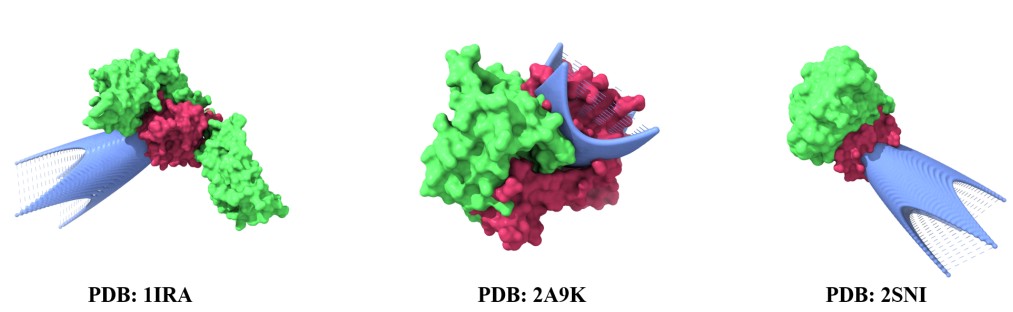

Figure 8: Visualization of elliptic paraboloid interfaces predicted by ElliDock on protein complex samples (from DB5.5). Red: ligand, green: receptor, blue: the predicted interface.

