# OpenReview forum: "Rigid Protein-Protein Docking via Equivariant Elliptic-Paraboloid Interface Prediction"
_ICLR.cc/2024/Conference — ICLR 2024 poster_

### Official Review · Reviewer_ptkL · 2023-10-27

**Soundness:** 3 good
**Presentation:** 2 fair
**Contribution:** 2 fair
**Rating:** 5
**Confidence:** 4

**Summary:**

The paper studies the problem of rigid-body protein-protein docking, which is important for drug design and protein engineering. The proposed method, ElliDock, models the protein-protein docking interface using an elliptic paraboloid approximation. By aligning the interfaces of two unbound proteins, ElliDock ensures roto-translation equivariance. The experimental results show the method outperforms other deep learning-based methods and achieves the fastest inference speed.

**Strengths:**

1. The paper addresses a significant real-world problem and offers a comprehensive discussion of the related work, to the best of my understanding.
2. The idea of using an elliptic paraboloid to approximate the docking interface is interesting. This paper thoroughly discusses the advantages of this paraboloid-centric method over previous point cloud registration techniques. Furthermore, the introduction of various loss terms to derive meaningful interfaces improves the rationale behind the proposed method.

**Weaknesses:**

1. The major concern about the method lies in the comparison with traditional template-based methods, e.g., HDOCK. As shown in Tables 1&2, HDOCK outperforms other methods by a significant margin, suggesting it may already effectively address the rigid-body docking challenge. Although the ElliDock’s inference speed is much faster than HDOCK, its performance is unsatisfactory. Given real-world applications, the value of faster inference time seems negligible, especially if the docking challenge doesn't necessitate high-throughput situations. Please correct me if I’m wrong.
2. Despite the considerable depth in the methods section discussing the rationale and procedure of using an elliptic paraboloid for approximation, it remains unclear how this looks like on real-world data. It would be good if there are some visualization experiments showing the learned paraboloid interfaces.
3. The paper's notations appear cluttered and confusing. For example, iEq.(8) denotes the other protein as $H_{-p}$, which is unconventional. In Eq. (12), the FC layer maps representation to $4$-dim without explanation. Meanwhile, Eq. (13) introduces the hyperparameter $M$ without elaboration. The bracket notation further complicates understanding. For better clarity, consider excluding the $p$, since most operations apply similarly to both proteins, and employ subscript notation for indexing.
4. A deeper exploration and connection with surface-based methodologies, such as MaSIF, would be beneficial. It would be insightful to include a comparison with it in the benchmarking section.

**Questions:**

1. In Eqs. (3)(4), the coefficient $\alpha_{j\to i}$ appears to merely normalize the inner product of $q$ and $k$, without taking into account the keys from other edges. This approach seems to limit the attention mechanism's capacity to capture interactions among edges or nodes. Given this, wouldn't it be more straightforward to directly map the message $m_{j\to i}$ to a scalar coefficient $\alpha_{j\to i}$?
2. In Sec. 4.3, there's a mention that pre-trained models like HDOCK could be prone to data leakage issues. From my understanding, HDOCK functions as a template-based model and doesn't utilize learning in its scoring function. Could you provide a more detailed explanation regarding the concerns of data leakage?
3. In Eq. (12), while $F_p$ is identified as a 4-dimensional vector, its fourth element seems not to be used in the main paper.

Overall, I appreciate the method introduced to address the rigid-body docking problem. I’m willing to raise my score if all the concerns listed in the Weakness section can be addressed during rebuttal.

---

> ### Author Response · Authors · 2023-11-21
>
> We sincerely thank your constructive comments. We address your concerns below accordingly.
>
> - The major concern about the method lies in the comparison with traditional template-based methods, e.g., HDOCK. As shown in Tables 1&2, HDOCK outperforms other methods by a significant margin, suggesting it may already effectively address the rigid-body docking challenge. Although the ElliDock’s inference speed is much faster than HDOCK, its performance is unsatisfactory. Given real-world applications, the value of faster inference time seems negligible, especially if the docking challenge doesn't necessitate high-throughput situations. Please correct me if I’m wrong.
>
> Thank you for the comment. While we respectfully agree with your point that HDOCK is a strong baseline in performance, we believe that models with faster inference time are more suitable in high-throughput scenarios. For example, in drug design, when studying whether inserting, editing or deleting sequence fragments in different regions of a protein affects the binding ability and binding site. If ElliDock is used in this case to simulate the binding process among the edited proteins, its fast speed will nicely support a large number of simulated docking processes within an acceptable time, such that we can locate the desirable protein candidate from its huge amino acid space.
>
> - The notation issues.
>
> Thank you very much for raising these issues. We have made corrections to the notations you mentioned. In Eq. (8), we use $\neg{p}$ to denote the other protein. In Eq. (12), the first three elements of $F_1$ and $F_2$ are used to construct the elliptical paraboloid, and the fourth element is used for rotation refinement, which are both mentioned in the paper. $M$ is a hyperparameter introduced in Eq. (13), which is further specified in the paragraph under Eq. (16). Regarding the problem of excluding $p$, we have formulas that only apply to a single protein (e.g., Eq. (8)), formulas that require information interaction (e.g., Eq. (7)), and formulas that require both proteins (e.g., Eq. (14)), so we believe that the subscript $p$ can make the expression clearer.
>
> - A deeper exploration and connection with surface-based methodologies, such as MaSIF, would be beneficial. It would be insightful to include a comparison with it in the benchmarking section.
>
> Thank you for introducing MaSIF as a surface-based methodology. From a methodological point of view, MaSIF models a real surface with biochemical properties, making use of protein surface chemical features such as hydropathy and free electrons; ElliDock models a fictional elliptical paraboloid as a surface, which does not represent chemical properties, yet represents the position information of the pockets as well. On the other hand, MaSIF can obtain its fingerprint from protein monomers, which is conducive to studying the properties of the monomers, while ElliDock interacts with the features of receptors and ligands during the docking process, which may improve the expressiveness on complexes.
>
> To address your concern, we have added the content to the Introduction section in the revised version.
>
> - Q1: about attention mechanism.
>
> We believe that the inner product of $q$ and $k$ has a larger representation space than direct mapping of the message $m$, because the former is a mapping of $\mathbb{R}^{H\times H}$ to $\mathbb{R}$, and the latter is a mapping of $\mathbb{R}^{H}$ to $\mathbb{R}$. Please correct me if I'm wrong.
>
> - Q2: the possible data leakage of HDock.
>
> Yes, we can show the evidence of the possible data leakage of HDock. From the paper of HDock [1], we find that HDock may have data leakage in two parts: first, during the docking procedure, HDock needs to search the whole PDB (thus may also include the testing data in our experiments) for homologous templates and build the 3D structures through homology modeling, and HDock will update the PDB data through version; second, the protein-protein scoring function [2] used by HDock, referred to as ITScore-PP, was derived using the crystal structures of a training set of 851 protein–protein dimeric complexes containing true biological interfaces.
>
> - Q3: about $F_p$.
>
> Sorry for the confusion. The fourth element of $F_p$ is used during rotation refinement, which is shown in Section 3.5.
>
> [1] Yan, Y., Tao, H., He, J., & Huang, S. Y. (2020). The HDOCK server for integrated protein–protein docking. Nature protocols, 15(5), 1829-1852.
>
> [2] Huang, S. Y., & Zou, X. (2008). An iterative knowledge‐based scoring function for protein–protein recognition. Proteins: Structure, Function, and Bioinformatics, 72(2), 557-579.

---

> > ### Author Response · Authors · 2023-11-22
> >
> > Dear reviewer, could you please give us a reply to our discussion? Thank you!

---

> > ### Comment · Reviewer_ptkL · 2023-11-22
> >
> > I've read the authors' response. As my questions about surface-based baselines and notations have not been addressed in the revised version, I would not raise my score.

---

> ### Author Response · Authors · 2023-11-22
>
> Sorry for the inconvenience that the revised part of the paper was not marked in red previously, now we have resubmitted our revised paper. For the surface-based baselines you mentioned, we have added the discussion of MaSIF in the Introduction section. We have also modified the notation issue. Meanwhile, we provide visualization of predicted elliptic paraboloid interfaces in the appendix. Please let us know if you have other concerns.

---

### Official Review · Reviewer_SQcb · 2023-10-30

**Soundness:** 3 good
**Presentation:** 3 good
**Contribution:** 3 good
**Rating:** 8
**Confidence:** 3

**Summary:**

The paper introduces a novel supervised learning approach for protein docking, focusing on predicting the interface between ligand and receptor proteins in the form of an elliptic-paraboloid. This method operates under the assumption that the true docked pockets are precisely situated along our predicted interface. The primary aim of this learning process is to identify elliptic-paraboloids that align with these ground-truth docked pockets. This objective is complemented by additional constraints, such as the loss associated with elliptic-paraboloid overlap, which ensures spatial separation of protein pairs at the interface, and the standard dock loss, quantifying the difference between predicted and ground-truth rotation and translation matrices.

The authors illustrate that the key advantage of this method lies in its efficiency during inference. Once the elliptic-paraboloid interface is predicted, it enables the efficient calculation of docking poses. Furthermore, this proposed method exhibits equivariance to arbitrary rotations and translations of the proteins, a crucial feature ensuring the generalization of the docking process.

To evaluate the performance of their approach, the authors conducted experiments using two specific benchmark datasets: "Docking benchmark version 5" and "The Structural Antibody Database." They compared their method to four baseline machine learning-based docking techniques: HDock (pretrained), Multimer (pretrained), EquiDock (trained from scratch), and DiffDock-PP (trained from scratch).

In the first benchmark dataset, the authors demonstrated that their ElliDock outperforms most of the baseline methods in terms of docking accuracy while achieving a significant reduction in inference time, with the exception of the pretrained model.

In the second benchmark dataset, the authors showed that ElliDock outperforms most of the baseline methods, except for HDock. The authors noted that HDock's superior performance may be attributed to the possibility of data leakage. They further conducted an ablation study to highlight the importance and effectiveness of each training objective function introduced in the ElliDock method.

**Strengths:**

The concept of streamlining the learning process through the introduction of an inductive bias that transforms a complex docking learning problem into the prediction of coinciding elliptic-paraboloids is truly interesting.

The experimental findings are particularly compelling, notably the notable enhancements in terms of inference efficiency. The experiments were conducted with great attention to detail, involving thorough comparisons with state-of-the-art machine learning-based docking methods.

The paper was thoughtfully composed, and it's commendable that the source code is available as open-source, making it accessible for others to explore and implement.

**Weaknesses:**

Challenging the assumption that elliptic-paraboloids precisely correspond to ground-truth docked pockets is a substantial step. To validate this assumption, it would be beneficial for the authors to consider conducting experiments using an established docking database, providing empirical evidence regarding the validity of this premise.

The results in comparison to models trained from scratch are indeed remarkable. While the authors suggest that the success of baseline models like HDock may be due to potential data leakage, it can be exceptionally challenging to definitively prove such claims. More compelling evidence supporting the presence of leakage in HDock would be valuable.

The authors demonstrate a significant improvement in inference time for their method compared to baseline approaches, yet the exact source of this efficiency enhancement remains somewhat unclear. An in-depth complexity analysis would be helpful in shedding light on the reasons behind this improvement. Typically, if a method directly predicts rotation and translation matrices, inference time should be as swift as the execution of these operations.

In addition to comparisons with machine learning-based docking methods, it would be beneficial to provide an overview of results in comparison to well-established docking methods that do not rely on machine learning techniques.

Furthermore, sharing the splits of the second benchmark dataset used in your experiments would be advantageous, as it would enable others to replicate the results more easily and conduct experiments with the same splits in the future. This transparency promotes reproducibility in the research community.

**Questions:**

Would you kindly offer substantiating evidence or theoretical findings to validate the assumption that elliptic-paraboloids accurately align with the ground-truth docked pockets?

    Could you furnish more compelling evidence to substantiate the assertion of potential data leakage in HDock, as stated in the paper?

    Can you provide a more comprehensive explanation regarding the reasons behind the improved inference time compared to other methods?

    Could you consider releasing the splits of the second dataset utilized in your experiments to facilitate reproducibility efforts within the research community?

---

> ### Author Response · Authors · 2023-11-21
>
> We sincerely thank your constructive comments. We address your concerns below accordingly.
>
> - In addition to comparisons with machine learning-based docking methods, it would be beneficial to provide an overview of results in comparison to well-established docking methods that do not rely on machine learning techniques.
>
> Thank you for the suggestion of adding baseline models that do not rely on machine learning techniques. In fact, as a template-based model, HDock is not strictly a machine learning model, and we choose HDock as a representative of the traditional docking method due to its state-of-the-art performance among traditional docking methods (based on the experiments of EquiDock [1]).
>
> - Would you kindly offer substantiating evidence or theoretical findings to validate the assumption that elliptic-paraboloids accurately align with the ground-truth docked pockets?
>
> Thanks! We provide more explanations on DB5.5 here. DB5.5 is a well established docking database. From our point of view, IRMSD measures the ability of the model to align to pockets. In Tables 1 and 2, the performance of our model on IRMSD is better than that of models trained from scratch, indicating that our method is better at finding pockets. We are open to discussing if you still have any questions.
>
> - Could you furnish more compelling evidence to substantiate the assertion of potential data leakage in HDock, as stated in the paper?
>
> Thank you for giving us the chance. From the paper of HDock [2], we find that HDock may have data leakage in two parts: first, during the docking procedure, HDock needs to search the whole PDB (thus may also include the testing data in our experiments) for homologous templates and build the 3D structures through homology modeling, and HDock will update the PDB data through version; second, the protein-protein scoring function [3] used by HDock, referred to as ITScore-PP, was derived using the crystal structures of a training set of 851 protein–protein dimeric complexes containing true biological interfaces.
>
> - Can you provide a more comprehensive explanation regarding the reasons behind the improved inference time compared to other methods?
>
> Thanks. Here we briefly analyze reasons for the differences between the inference time of different models. First of all, ElliDock and EquiDock are both regression models and generate the transformation in one step from end to end. Therefore, there is no order of magnitude difference between them in speed. From Table 1, we show ElliDock is $\textasciitilde$2x faster than EquiDock, which is more likely due to the amount of model parameters, number of model layers, etc.. DiffDock-PP is a diffusion-based model, with a denoise process of T steps (T is a hyperparameter) in the inference procedure. Each inference step requires re-running the model, so there is a speed gap of 10$\textasciitilde$100 times compared with the regression model. HDock is a template-based model that not only obtain the templates of receptors and ligands through sequence alignment, but also needs to evaluate the energy of the predicted complex based on the scoring function and perform multi-step updates accordingly. Meanwhile, HDock can only run on the CPU, limiting its speed even more. Multimer is a deep learning model, yet it requires multiple sequence alignment (MSA) during training and inference, and its pre-trained models have a large number of parameters (GB level), so the inference time is extremely long.
>
> - Could you consider releasing the splits of the second dataset utilized in your experiments to facilitate reproducibility efforts within the research community?
>
> We have already uploaded the bash file for downloading and spliting the dataset. Please check README in our source code for more details. All the code sources will be released after our paper is accepted.
>
> [1] Ganea, O. E., Huang, X., Bunne, C., Bian, Y., Barzilay, R., Jaakkola, T., & Krause, A. (2021). Independent se (3)-equivariant models for end-to-end rigid protein docking. arXiv preprint arXiv:2111.07786.
>
> [2] Yan, Y., Tao, H., He, J., & Huang, S. Y. (2020). The HDOCK server for integrated protein–protein docking. Nature protocols, 15(5), 1829-1852.
>
> [3] Huang, S. Y., & Zou, X. (2008). An iterative knowledge‐based scoring function for protein–protein recognition. Proteins: Structure, Function, and Bioinformatics, 72(2), 557-579.

---

> > ### Comment · Reviewer_SQcb · 2023-11-22
> > **Thank you for the response**
> >
> > Dear authors,
> > Thank you for the responses!
> > I have read the other reviewer's comments and your responses. I am happy with the responses and the reproducibility of the proposed approach.
> > Best regards,

---

> > > ### Author Response · Authors · 2023-11-22
> > >
> > > Thank you for your recognition of our work! Best regards.

---

### Official Review · Reviewer_5pax · 2023-11-01

**Soundness:** 3 good
**Presentation:** 3 good
**Contribution:** 3 good
**Rating:** 8
**Confidence:** 3

**Summary:**

This paper proposes a rigid protein-protein docking model, ElliDock, that functions by predicting a pair of paraboloids on the ligand and receptor with the same shape and aligning them via roto-translation such that they coincide. The paper's main competitors are EquiDock, DiffDock-PP, and AlphaFold-Multimer, all of which rely on different methods. ElliDock outperforms all of these models in terms of inference time. It outperforms EquiDock and DiffDock-PP on the DB5.5 dataset, and all models on the SAbDab dataset.

**Strengths:**

1. The is high-quality and presented very well.
2. The argument as to why EquiDock suffers compared to ElliDock is developed well.
3. The method paraboloid matching in the context of protein-protein docking is seemingly novel and mathematically developed very well in the paper.
4. The results are convincing.

**Weaknesses:**

1. The paraboloid formulation is elegant, but I would like to see some discussion as to why this is the right choice compared to other considered options.
2. The argument of why ElliDock's formulation is better than EquiDock seems to be developed sufficiently well, but I would like to see some more discussion contrasting with DiffDock-PP and AlphaFold-Multimer.

**Questions:**

1. The inference time is reported in Table 1, but not in Table 2. Could you please report the values for Table 2?
2. Could you please report the inference time in Table 3. It would be interesting to see which losses could be removed for even more speed-up, but without accuracy loss.
3. Since DiffDock-PP is a diffusion-based model it is capable of modelling multiple conformations. How does ElliDock fare when the ligand-protein pair has multiple possible conformations?
4. Could you argue a bit about why ElliDock's training objective is more suitable than DiffDock-PP's?
5. Why do you think ElliDock outperforms AlphaFold-Multimer on SAbDab?

---

> ### Author Response · Authors · 2023-11-21
>
> We sincerely thank your constructive comments. We address your concerns below accordingly.
>
> - The paraboloid formulation is elegant, but I would like to see some discussion as to why this is the right choice compared to other considered options.
>
> Thank you for recognizing that our work. We choose elliptic paraboloids other than planes, spheres or complex manifolds to model the binding interface, as elliptic paraboloids are appropriately defined in 3D space and their SE(3) transformations can be devised in a closed form. On the one hand, if using planes, spheres or other symmetric 2D manifolds to model the interface, it will easily cause the docking ambiguity issue since there are usually different ways to transform a 2D manifold from one pose to another.  On the other hand, if using complex manifolds, it may introduce bias and higher computational complexity to obtain the docking solutions.  Under these two requirements, we believe that the elliptical paraboloid is a good trade-off.
>
> - The argument of why ElliDock's formulation is better than EquiDock seems to be developed sufficiently well, but I would like to see some more discussion contrasting with DiffDock-PP and AlphaFold-Multimer.
>
> Sorry for the insufficient explanations. We know that our work ElliDock and EquiDock are both regression models, which allows an easier comparison between two methods. On the contrary, DiffDock-PP is a diffusion-based model, it successfully builds a model that meets the degree of freedom required by the task. The introduction of noise in diffusion steps gives DiffDock-PP the ability to find multiple binding pockets, while it also leads to 10~100 times of the inference time compared to regression models. Alphafold-Multimer is based on Alphafold, its task setting is the structure prediction from sequence instead of the rigid protein-protein docking, thus the method design is different from other methods mentioned above.
>
> - The inference time is reported in Table 1, but not in Table 2. Could you please report the values for Table 2?
>
> Nice suggestion! We have additionally updated the inference time in Table 2 in the revision. We believe that the inference time should be approximately proportional to the scale of the test set, so we only reported the results of one test set previously.
>
> - Could you please report the inference time in Table 3. It would be interesting to see which losses could be removed for even more speed-up, but without accuracy loss.
>
> Thank you for the comment. We would like to clarify that the losses only play a role in the training process and have little impact on the inference time during validation and testing. Hence, it is not very informative to report the inference time in Table 3 (the time is almost the same no matter which loss we remove).
>
> - Since DiffDock-PP is a diffusion-based model it is capable of modelling multiple conformations. How does ElliDock fare when the ligand-protein pair has multiple possible conformations?
>
> Thank you for raising this question. Like EquiDock and other regression-based methods, our Ellidock is not specifically designed to handle the situation when there are multiple binding sites for one single pair of receptor and ligand. However, for the datasets, particularly the antibody-antigen binding tasks, the specificity of the antibody is almost only determined by a small special region of itself (complementarity determining region, CDRs). In actual scenarios, it might be difficult for the same pair of antibody and antigen to have multiple binding sites.
>
> - Could you argue a bit about why ElliDock's training objective is more suitable than DiffDock-PP's?
>
> In our perspective, since ElliDock and DiffDock-PP are different types of models, the training objectives are difficult to compare directly. The training target of DiffDock-PP is mainly the score function of rotation and translation, which pays more attention to the distribution changes of the overall molecule. ElliDock is based on interface fitting and tends to focus more on pockets in the design of training objectives.
>
> - Why do you think ElliDock outperforms AlphaFold-Multimer on SAbDab?
>
> Thanks. Considering only the SAbDab data set, since the pre-training data of Multimer comes from the general protein domain, it probably performs poorly on the antibody-antigen domain. In Table 2 we show that ElliDock has better performance on both CRMSD and IRMSD metrics. As for the DockQ metric, it can be seen from Figure 5 that ElliDock has better prediction results on most test samples. However, Multimer has achieved extremely amazing accuracy on a few samples, which may be attributed to the powerful representation ability of Alphafold.

---

> > ### Comment · Reviewer_5pax · 2023-11-22
> >
> > Thank you for the reply. I have no further comments.

---

> > > ### Author Response · Authors · 2023-11-22
> > >
> > > Thank you for your recognition of our work! Best regards.

---

### Official Review · Reviewer_raHF · 2023-11-13

**Soundness:** 3 good
**Presentation:** 3 good
**Contribution:** 2 fair
**Rating:** 3
**Confidence:** 4

**Summary:**

This paper proposed a deep learning method called ElliDock on solving rigid proteins docking. Firstly, they use a pair of elliptic paraboloids to estimate docking interfaces and then use equivariant neural network to learn rotation and translation. The first part is interesting. They compared with classical docking method HDock and deep learning based methods, EquiDock, DiffDock-PP, multimer. Experimental results show some minor improvement on these old deep learning baselines and much worse than HDock. My major concern are: 1) they didn't compare with the latest method, 2) the setting is not realistic. see weaknesses i listed as follow.

**Strengths:**

- this paper predicted a pair of elliptical paraboloids as two proteins' binding interfaces. This idea is interesting.

- this method compared both classical docking method, e.g., HDock and deep-learning based docking method, e.g., EquiDock, Multimer.

- it's glade to see experimental comparison on both general protein complex and antibody-antigen complexes. The last one is important in the field of drug design.

**Weaknesses:**

- so small font size in figures, for example, Figure 1.

- the improvement is minor on DB5.5 when comparing with EquiDock and DiffDock-PP and much worse than Multimer and HDock.

- [1] and [2] are stronger baselines on solving antibody-antigen complex.

[1] Yujie Luo, et.al. xTrimoDock: Rigid Protein Docking via Cross-Modal Representation Learning and Spectral Algorithm.

[2] Ambrosetti, Francesco, et.al. Modeling antibody-antigen complexes by information-driven docking.

- This paper claims ElliDock solves steric clashes well. I guess one reason is this experiment uses native unbound structure as the input. There is a trade-off between steric clashes and dockq if you use predicted structures. When using predicted structures, there is no perfect pocket to fit ligand on receptor.

**Questions:**

- HDock performs much better than ElliDock. I guess the major reason is the input rigid structure is native unbounded structure. However, using native unbound structure as docking method's input is not realistic. What's the performance if you dock predicted structures.

- the pair of elliptical paraboloids are with the same shape. How to handle large protein or samll proteins?

---

> ### Author Response · Authors · 2023-11-21
>
> We sincerely thank your constructive comments. We address your concerns below accordingly.
>
> - so small font size in figures, for example, Figure 1.
>
> Thank you for your suggestions on fonts. We have adjusted the font sizes of some figures.
>
> - the improvement is minor on DB5.5 when comparing with EquiDock and DiffDock-PP and much worse than Multimer and HDock.
> We would like to further emphasize the advantages of our Ellidock.
>
> First, as mentioned in the paper, we use the pre-trained models of Multimer and HDock, both of which use the data in the PDB for training (Multimer) or template selection (HDock). Since DB5.5 is included in PDB, we assume that Multimer and HDock perform extremely well on DB5.5 due to possible data leakage. The small scale of DB5.5 might also limit the learning ability of the model.
>
> Second, inference time is another important factor in addition to the performance metric. Clearly, ElliDock is much faster than HDock, DiffDock-PP, and Multimer. In our responses to other reviewers, we believe that models with faster inference time are more suitable in high-throughput scenarios. For example, in drug design, when studying whether inserting, editing or deleting sequence fragments in different regions of a protein affects the binding ability and binding site. If ElliDock is used in this case to simulate the binding process between the editted proteins, its fast speed will nicely support a large number of sumulated docking processes within an acceptable time, such that we can locate the desirable protein candidate from its huge amino acid space.
>
> - [1] and [2] are stronger baselines on solving antibody-antigen complex.
>
> Thank you for introducing two baselines for antibody-antigen complex. Unfortunately, the codes of both baselines have not been published, it is difficult for us to use them as experimental benchmarks. We will add further discussion in the revised paper.
> - HDock performs much better than ElliDock. I guess the major reason is the input rigid structure is native unbounded structure. However, using native unbound structure as docking method's input is not realistic. What's the performance if you dock predicted structures.
>
> While we also respect HDock for its promising performance, we have some concerns about its possible data leakage, as already expressed in our paper. From the paper of HDock [3], we find that HDock may have data leakage in two parts: first, during the docking procedure, HDock needs to search the whole PDB (thus may also include the testing data in our experiments) for homologous templates and build the 3D structures through homology modeling, and HDock will update the PDB data through version; second, the protein-protein scoring function [4] used by HDock, referred to as ITScore-PP, was derived using the crystal structures of a training set of 851 protein–protein dimeric complexes containing true biological interfaces.
>
> - The pair of elliptical paraboloids have the same shape. How to handle large proteins or small proteins?
>
> There could be some misunderstanding here. For different protein pairs, due to the model design of global information interaction, the shape of the elliptical paraboloid is determined by both the receptor and the ligand, so the interface shape can be adapted to the scale of the protein.
>
> [1] Yujie Luo, et.al. xTrimoDock: Rigid Protein Docking via Cross-Modal Representation Learning and Spectral Algorithm.
>
> [2] Ambrosetti, Francesco, et.al. Modeling antibody-antigen complexes by information-driven docking.
>
> [3] Yan, Y., Tao, H., He, J., & Huang, S. Y. (2020). The HDOCK server for integrated protein–protein docking. Nature protocols, 15(5), 1829-1852.
>
> [4] Huang, S. Y., & Zou, X. (2008). An iterative knowledge‐based scoring function for protein–protein recognition. Proteins: Structure, Function, and Bioinformatics, 72(2), 557-579.

---

> > ### Author Response · Authors · 2023-11-22
> >
> > Dear reviewer, could you please give us a reply to our discussion? Thank you!

---

### Author Response · Authors · 2023-11-21
**General response**

We thank all reviewers for their valuable comments. We have carefully revised our paper to address concerns listed below.
- More compelling evidence to substantiate the assertion of potential data leakage in HDock. (To Reviewer SQcb)
- Further explanations of how faster inference time matters in real world scenerios. (To Reviewer ptkL)
- Further explanations of choosing elliptic paraboloids as the interface compared to other considered options. (To Reviewer 5pax)
- Expanded discussions on related works and additional citations. (To Reviewer ptkL and Reviewer raHF)

---

### Meta-Review · Area_Chair_aaPR · 2023-12-08

**Metareview:**

Summary:

This paper proposes a rigid protein-protein docking model, ElliDock, that functions by predicting a pair of paraboloids on the ligand and receptor with the same shape and aligning them via roto-translation such that they coincide. The paper's main competitors are EquiDock, DiffDock-PP, and AlphaFold-Multimer, all of which rely on different methods. ElliDock outperforms all of these models in terms of inference time. It outperforms EquiDock and DiffDock-PP on the DB5.5 dataset, and all models on the SAbDab dataset.

Strengths:

- The paper uses a pair of elliptical paraboloids as two proteins' binding interfaces. This idea is interesting.
- This method compared both classical docking method, e.g., HDock and deep-learning based docking method, e.g., EquiDock, Multimer.
- Experimental comparison on both general protein complex and antibody-antigen complexes. The last one is important in the field of drug design.
- High-quality and presented very well.
- The argument as to why EquiDock suffers compared to ElliDock is developed well.
- The method paraboloid matching in the context of protein-protein docking is seemingly novel and mathematically developed very well in the paper.
- The results are convincing.
- Streamlining the learning process through the introduction of an inductive bias that transforms a complex docking learning problem into the prediction of coinciding elliptic-paraboloids is truly interesting.
- The experimental findings are particularly compelling, notably the notable enhancements in terms of inference efficiency.
- The experiments were conducted with great attention to detail, involving thorough comparisons with state-of-the-art machine learning-based docking methods.
- The paper was thoughtfully composed, and it's commendable that the source code is available as open-source, making it accessible for others to explore and implement.
- The paper addresses a significant real-world problem and offers a comprehensive discussion of the related work, to the best of my understanding.
- This paper thoroughly discusses the advantages of this paraboloid-centric method over previous point cloud registration techniques.
- The introduction of various loss terms to derive meaningful interfaces improves the rationale behind the proposed method.

Weaknesses:

- The paraboloid formulation is elegant, but I would like to see some discussion as to why this is the right choice compared to other considered options.
- I would like to see some more discussion contrasting with DiffDock-PP and AlphaFold-Multimer.
- Challenging the assumption that elliptic-paraboloids precisely correspond to ground-truth docked pockets is a substantial step. To validate this assumption, it would be beneficial for the authors to consider conducting experiments using an established docking database, providing empirical evidence regarding the validity of this premise.
- While the authors suggest that the success of baseline models like HDock may be due to potential data leakage, it can be exceptionally challenging to definitively prove such claims. More compelling evidence supporting the presence of leakage in HDock would be valuable.
- The exact source of efficiency enhancement remains somewhat unclear. An in-depth complexity analysis would be helpful in shedding light.
- It would be beneficial to provide an overview of results in comparison to well-established docking methods that do not rely on machine learning techniques.
- The major concern about the method lies in the comparison with traditional template-based methods, e.g., HDOCK. Although the ElliDock’s inference speed is much faster than HDOCK, its performance is unsatisfactory. Given real-world applications, the value of faster inference time seems negligible, especially if the docking challenge doesn't necessitate high-throughput situations.
- It remains unclear how this looks like on real-world data. It would be good if there are some visualization experiments showing the learned paraboloid interfaces.
- A deeper exploration and connection with surface-based methodologies, such as MaSIF, would be beneficial. It would be insightful to include a comparison with it in the benchmarking section.

Recommendation:

This is a borderline paper with some reviewers voting for rejection and other for acceptance. The most negative reviewer is of low quality and does not justify rejection with enought arguments. I, therefore, recommend accepting the paper and encourage the authors to use the feedback provided to improve the paper for the camera ready version.

**Justification For Why Not Higher Score:**

Some reviewers a leaning towards rejection.

**Justification For Why Not Lower Score:**

The most negative reviewer is of low quality and does not justify rejection well enough. The other remaining negative reviewer only leads slightly towards rejection while the other positive reviewers support strongly acceptance.

---

### Decision · Program_Chairs · 2024-01-16

Accept (poster)